# To develop online platform and determine its effectiveness in ENHANCING DIABetes knowledge among diabetes patients in primary CARE clinic (Enhancing-Diab-Care Study): Study protocol

**Hui Zhu Thew**[1]*, **Ai Theng Cheong**[1], **Sazlina Shariff Ghazali**[1], **Poh Ying Lim**[2], **Aneesa Abdul Rashid**[1], **Ping Foo Wong**[3], **Yong Kui Choon**[3], **Adam Firdaus bin Dahlan**[3], **Chitra Suluraju**[3], **Nadiah Othman**[3], **Yong Wai Leng**[3], **Izwan Zuhrin bin Abdul Malek**[4], **Yoke Mun Chan**[5], **Yan Bin Fong**[6], **Chee Han Ng**[6], **Ja Shen Choy**[6], **Shen Horng Chong**[6]

**1** Department of Family Medicine, University Putra Malaysia, Serdang, Selangor, Malaysia, **2** Department of Community Health, Universiti Putra Malaysia, Serdang, Selangor, Malaysia, **3** Primary Health Care Clinics, Ministry of Health Malaysia, **4** Department of Rehabilitation, Faculty of Medicine and Health Sciences, Universiti Putra Malaysia, **5** Department of Dietician, Faculty of Medicine and Health Sciences, Universiti Putra Malaysia, **6** Faculty of Medicine and Health Sciences, Universiti Putra Malaysia, Serdang, Selangor, Malaysia

* jothew@upm.edu.my

## Abstract

Diabetes Mellitus (DM)- Type 2 is increasingly prevalent in today's population. DM remains poorly controlled due to insufficient knowledge and understanding of this disease. Maintaining good diabetic self-care requires knowledge and empowerment. The COVID-19 pandemic had caused a shift of some public attention from non-communicable to communicable diseases, leading to patients becoming ignorant of their diabetic status. Therefore, we aim to develop an online education video platform that enables patients to enhance their knowledge about Diabetes Mellitus Type 2.

In this blocked randomised control trial, a minimum number of 232 participants with type 2 diabetes mellitus will be needed. Patients will consent and voluntarily participate during their follow-up at public primary health care clinic (Klinik Keshihatan Cheras Baru). Patients undergoing diabetic follow-up will be divided into two groups where only patients within the intervention group will receive diabetic care video. Patients' clinical profiles such as date of birth, gender, education status, Diabetes Knowledge Test (DKT) and Diabetic Empowerment Scale (DES-28) were collected to assess diabetes self-care knowledge and empowerment using self-administered questionnaires. Patients will be reminded to complete the educational video at the 3-month and 6-month follow-ups, and the aforementioned parameters will be reassessed. The data will be assessed using an independent t-test for the difference

**Data availability statement:** No datasets were generated or analysed during the current study. All relevant data from this study will be made available upon study completion.

**Funding:** GP-IPM/2022/9714000 funded RM30,000 for video production and publication. The sponsor (Inisiatif Putra Muda) was not involved in study design, data collection and analysis, decision to publish, or preparation of the manuscript.

**Competing interests:** The authors have declared that no competing interests exist.

between intervention and control groups. A Paired t-test will assess the difference between the patient pre and post-intervention after 6 months. A generalised Estimating Equation will be used to investigate the effectiveness of diabetes knowledge and clinical outcome, adjusted with covariates. $P < 0.05$ will be considered statistically significant. Ethical principles outlined in the Declaration of Helsinki and Malaysian Good Clinical Practice Guideline will be followed. Ethical approval will be obtained from MREC and NMRR before starting any study-related activities.

## Introduction

Diabetes is a growing global health concern, with its highest prevalence found in low- and middle-income countries and accounts for 1.5 million deaths per year. In Southeast Asia, type 2 diabetes is now the most prevalent disease. [1] According to the International Diabetes Federation (IDF), 34 million people in south-east Asia have diabetes. According to estimates, the number will rise to over 55 million by 2035. [1] In Malaysia, approximately 3.9 million people have diabetes, according to the National Health and Morbidity Survey 2019. This rate has increased from 13.4% in 2015 to 18.3% in 2019. Alternatively, that is approximately one-fifth of the adult population in Malaysia. [2] There is a high prevalence of poor glycaemic control with 7.9% mean HbA1c levels. [2] Several studies show that Malaysians with poor glucose control engage in poor self-care practices and have a limited understanding of diabetes. [3–6] Additionally, there are also insufficient diabetes educators to deal with a large number of diabetes patients. [7] Furthermore, COVID-19 has significantly impacted the entire healthcare system in recent years. Primary care and hospital practice are currently overburdened with diagnostic testing, monitoring, and management of COVID-19 care, resulting in routine maintenance for non-communicable diseases becoming compromised. [8] Given these factors, there is an urgent need for practical solutions.

Studies have demonstrated that diabetes education improves self-care and glucose control among people with low health literacy. At the same time, clinicians have less time and resources to disseminate information. [9,10] The number of diabetic education classes is disappointingly low, particularly among patients with lower socioeconomic classes, those who have not yet developed diabetes complications, and during the COVID-19 pandemic when most patients are unwilling to stay in the hospital for longer.

Diabetes self-care programs include health education, a critical component for managing and treating chronic diseases. [11] This principle underpins the theory that knowledge, awareness, and practices (KAP) of patients with diabetes are considered the most important factors for assessing health education outcomes. As the KAP theory outlines, human behaviour change can be divided into three steps: acquiring knowledge, creating attitudes and beliefs, and changing behaviours, during which human health behaviours can also be transformed. [12,13] The use of knowledge can assist patients in making informed decisions about their health

when dealing with non-communicable diseases such as diabetes. By empowering them, patients can increase their self-efficacy and improve their self-confidence. Utilising the media in publicity campaigns, creating educational materials, and educating people with diabetes and their families should be part of this effort. [14] Diabetes education and awareness programs will benefit healthy subjects as prevention measures and help people with diabetes better control their condition. [15,16] Providing specific and practical guidance for maintaining NCDs' critical health and community services is essential. National guidance is required to develop and use digital health solutions in NCD care, self-care, at-home care, and peer support. Yet, with the challenges of the COVID-19 pandemic facing us in the 21st century, it is vital to evaluate how medical information is delivered effectively through health promotion in the practice of new norms. The systematic reviews on using technology to facilitate diabetes self-management reported a significant reduction in glycated haemoglobin and cost-effectiveness. With internet-delivered diabetes education, many individuals have easier access to the material and can take their time with the learning process. [17–19]

Therefore, this work aims to develop and investigate the effectiveness of web-based education videos that include tailored information on knowledge to educate type 2 diabetes patients in primary care.

## Materials and methods

### Study design

The study design used in this study is a Blocked Randomized Control Trial (RCT) ratio one to one. Participants will be randomly assigned in a 1:1 ratio to either the treatment or control group to minimise biases in outcome assessment and data analysis. The data collector will remain blinded to group assignments and undergo training to adhere to standardized protocols during data collection. Additionally, random monitoring of data collection will be implemented to ensure and sustain reliability. Inclusion and exclusion criteria have been designed to control for confounding variables. The study will be conducted and evaluated by the requirement in the Consolidated Standards of Reporting Trials (CONSORT) Statement. [20] Before initiating any study-related activities, ethical approval will be obtained from Medical Research & Ethics Committee (MREC) and National Medical Research Registry (NMRR). Our research study was registered under Thai Clinical Trials Registry (identification number: TCTR20230321005). Written consent will be taken from the participants. They will receive patient information sheets in English or Malay based on preference, with a one-week ample period for informed decision-making. The principal investigator can be contacted if any clarifications are needed.

### Study population

The study population consists of type 2 diabetes patients who registered in walk-in and follow-up clinics in the primary care clinics (Klinik Kesihatan Cheras Baru) from November 1, 2023, to May 31, 2024.

### Study phase

**Pre-development phase.** An initial literature review was conducted as part of this preliminary phase to scope out the systematic reviews and guidelines of the type 2 diabetes education online education module. In total, 12 topics were identified by the primary care physician and three patients as essential diabetes knowledge. According to our research, there are no standard diabetes education videos available online in English, Malay, Chinese, and Tamil for healthcare practitioners dealing with type 2 diabetes. (Fig 1)

**Development phase.** The main video content is in English, and the script is based on the 6th version Clinical Practice Guideline - Management of Type II Diabetes Mellitus [21], alongside references from the American Association of Clinical Endocrinology (AACE) and the European Association for the Study of Diabetes (EASD). The information will be summarised and written in simpler for easier understanding. A set of English-version online education videos will be developed in this phase. They will be presented in a simple, short, and straightforward manner to facilitate patients' understanding of self-care for type 2 diabetes. Each video is intended to last no more than ten minutes.

Figure. RCT enrollment, intervention and assessments.

| | STUDY PERIOD | | | | |
| --- | --- | --- | --- | --- | --- |
| | Enrolment | Allocation | Post-allocation | | Close-out |
| TIMEPOINT | $-t_1$ | 0 | $t_1$ (3 months post-allocation) | $t_2$ (6 months post-allocation) | $t_x$ |
| **ENROLMENT:** | | | | | |
| **Eligibility screen** | X | | | | |
| **Informed consent** | X | | | | |
| **Allocation** | | X | | | |
| **INTERVENTIONS:** | | | | | |
| *Web-based educational videos (in intervention group only)* | | | X | X | |
| *3-monthly follow up and consultation (in both intervention and control group)* | | | X | X | |
| *Video education (in both intervention and control group)* | | | | | X |
| **ASSESSMENTS:** | | | | | |
| *Anthropometric and clinical profile data* | | X | X | X | |
| *Diabetes knowledge and empowerment score data* | | X | X | X | |

**Fig 1. Schedule of enrolment, intervention, and assessment.** Schedule of enrolment, interventions, and assessments according to SPIRIT protocol tools.

The videos will be created from the ground up. All branding logos will be carefully designed to respect intellectual property rights, and the graphics and background music will be sourced from free, authorised channels to avoid copyright concerns. The logos will also be custom-designed to ensure originality.

Upon completion of recording and editing, each version of the videos in the four languages (English, Malay, Chinese, and Tamil) will undergo the copyright process to protect their intellectual property.

**Review and finalisation phase.** Following the completion of the scripts, all experts and patients were invited to participate in the review and finalisation phase. The expert panel comprises two Family Medicine Specialists, an

Ophthalmologist, a Pharmacist, two Endocrinologists, a Public Health Specialist, a Rehabilitation Specialist, a Dietician, and a Chinese, a Malay, and an Indian patient. These were two different groups of patients, and the patients were different and did not overlap with the trial patients. They will review the scripts and complete a questionnaire based on the 12 video topics. Each topic will be evaluated on two items: relevance and clarity, using a 3-point scale, modified from the content validity tool. [22] where 1 indicates 'not relevant' or 'not clear,' 2 indicates 'somewhat relevant' or 'somewhat clear,' and 3 indicates 'highly relevant' or 'highly clear.' The content and validity of each video were acceptable, with a score of 0.8. The English version was translated into Malay, Chinese, and Tamil languages by the same group of author, and a pilot study was conducted. A sample of 5 respondents for each language was selected. Some minor grammatical errors and difficult terminology were identified and addressed accordingly. A reliability test was conducted again for the Malay, Chinese, and Tamil languages, and the results were acceptable.

**Pilot testing phase.** Each language version of the final videos will be tested with ten patients for this phase face-to-face, and their feedback will be solicited. Documentation of the sessions is being conducted to understand better how patients perceive the videos' usefulness and ease of use.

## Data collection

The researchers will collect data by selecting type 2 diabetes patients from the baseline. Only patients with type 2 diabetes who meet all inclusion criteria and do not satisfy any exclusion criteria will be recruited in our study. A research assistant will approach patients on their follow-up visit at Klinik Kesihatan Cheras Baru. As part of the consent process, the investigator will explain and provide the Patient Information Sheet to the participant and an agreement to receive calls and messages from the researcher for research purposes. They will keep their phone numbers private and only be contacted or messaged every four weeks to remind them to complete the video-watching and follow-up appointment. The patients will, however, be given one week to consider their participation if they are still deciding on the recruitment day. We will conduct a one-to-one ratio randomisation control trial with the control and intervention groups.

During the same day, all participants will be given a hard copy self-administered questionnaire by research assistance to complete, and their latest clinical profile (height, weight, BMI, waist circumference, blood pressure, latest blood test for HbA1c and fasting lipid profiles) will be taken from their medical records. The self-administered questionnaire will take about 20 mins to complete. This stage will be considered a baseline assessment for control and intervention groups. After the questionnaire, both groups will follow up in the clinic as usual. The intervention group will take place after all the participants complete the questionnaire. The participants will be told about the structure of the program. There will be research assistants the researcher has already trained to assist and teach the participants how to assess and use the web-based online education videos with a QR code. They will need to watch all the videos during the study period in their free time. The research assistant will follow up with their progress and remind them to complete the video every four weeks via phone call. After three months of the intervention, both the intervention and control group participants will return to the clinics for follow-up. During the follow-up, participants will be given a self-administered questionnaire to answer.

Six months from the baseline, the intervention and control group participants will return to the clinics for follow-up. During the follow-up, participants will be given a self-administered questionnaire to answer, and their latest clinical profile (height, weight, BMI, waist circumference, blood pressure, latest blood test for HbA1c and fasting lipid profiles) will be taken from their medical records. As a result of this study, participants will watch a video to learn about diabetes and how to take care of themselves; no further treatment will be administered after the study is completed. Patients who wish to withdraw from the study can withdraw anytime by informing the principal investigator. The flow of the process is summarised in (Fig 2).

## Inclusion/Exclusion criteria

The inclusion criteria of participants are as follows: 1) type 2 diabetes patients aged 18 years or older, 2) have been diagnosed and followed up for type 2 diabetes for at least six months, 3) have an HbA1c greater than 8% at the time of

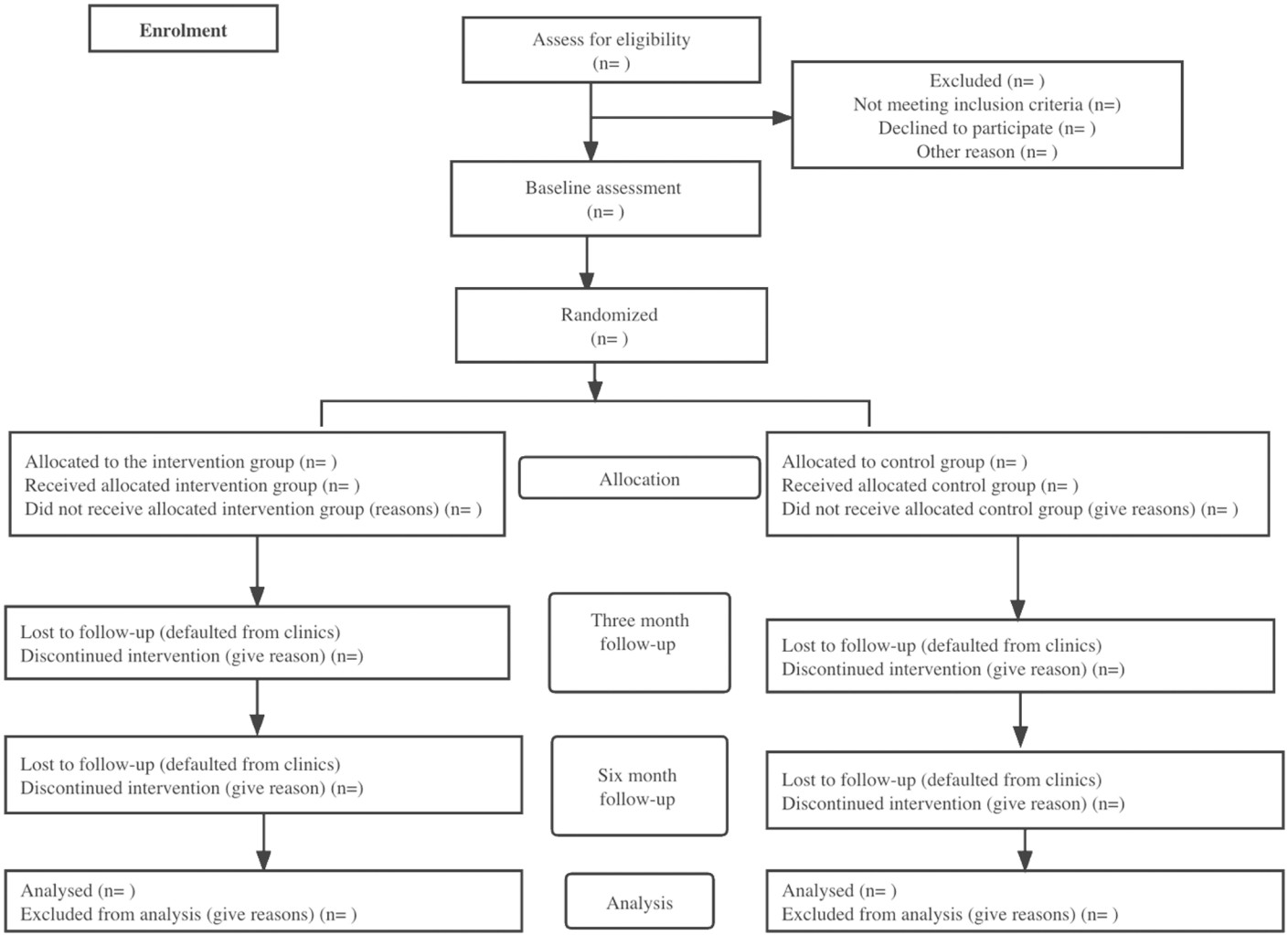

**Fig 2. Study flowchart.** Flow of the process from enrolment to baseline assessment, randomization, follow-ups and data analysis.

enrolment, 4) only one family member from each household will participate in the study, 5) able to understand Malay, 6) have internet access at home.

Exclusion criteria are as below: 1) patients who are acutely ill, 2) have psychiatric illness, 3) bedridden and rely on nursing care to perform daily activities, 3) intellectually disabled, as determined by a community-centred board.

## Study instrument

As a baseline measure, respondents will be provided with self-administered questionnaires in the Malay language upon initiation of recruitment. This questionnaire will have three sections: Section one will cover sociodemographic factors (age, gender, ethnicity, education level, marital status, employment status, income, smoking status, and history of hospitalisation for complications related to type 2 diabetes). The second section consists of a diabetes knowledge test (DKT) which is a validated test for assessing the knowledge of a patient about diabetes, and the third consists of a diabetes empowerment score (DES-28) questionnaire which is also a validated questionnaire to assess the psychosocial self-efficacy of diabetic patients. For the translated questionnaire, Cronbach's alpha was acceptable (α=0.573). The questionnaire was

administered in Malay because Malay is the official language in Malaysia. The main reason for creating the video in four languages (Malay, Chinese, Tamil, and English) is due to the multiracial background of the patients. Learning in one's own mother tongue has been proven to be more effective [23].

A series of web-based educational videos will be developed by primary care physicians, dietitians, physiotherapists, occupational therapists, and dentists before assessing the general knowledge of diabetes and diabetes self-care among our respondents. After the English version, the videos will be translated into Malay, Chinese and Tamil. These videos will cover the following topics: 1) An Introduction to Enhancing Diabetes Care. 2) A guide to screening, diagnosis and HbA1c test. 3) Oral anti-diabetes medications. 4) Injection form anti-diabetes medications and injection technique. 5) Self Monitoring Blood Glucose (SMBG). 6) Hypoglycemia. 7) Dietary. 8) Self-care. 9) Exercise. 10) Understanding complications and prevention. 11) Recap and Roadmap to a better future. 12) Real-life journey: A type 2 diabetes patient shares his testimony. Before the video recording, the expert panellists will review, edit, and validate the videos using a video validation questionnaire with input from endocrinologists, primary care physicians, pharmacists, ophthalmologists, dietitians, rehabilitation specialists, and a few patients.

## Sample size

The sample size was calculated based on two mean groups formula [24]; mean±sd of knowledge differences for the intervention group and control group were 0.7±2.21 and -0.30±2.33 [17], 80% of power, 5% of risk error, 30% attrition rate, the sample size needed for each group is 116, with the total is 232.
Two mean formula is stated as below:

$$n = \frac{2\sigma^2 \left[ z_{1-\frac{\alpha}{2}} + z_{1-\beta} \right]^2}{(\mu_1 - \mu_2)^2}$$

N sample size estimate
Z1-α/2 standard error with 95% confidence interval = 1.96
z1−β Standard error associated with 80% power = 0.842
μ_1 mean of knowledge score in intervention group
μ_2 mean of knowledge score in control group
Estimated standard deviation

$$n = \frac{2(2.27)^2 [1.96 + 0.842]^2}{(1)^2}$$

=81 per group
Adjusted with 30% attrition rate = 81/0.7
=116 per group
Total sample size is 116 x 2 = 232

## Statistical analysis plan

In this study, the primary outcome of interest is diabetes knowledge, and the secondary outcome is the clinical outcome (HbA1c, blood pressure, fasting lipid profile, BMI, waist circumference) among intervention and control groups at baseline. The Independent variable will be the web-based education video for the intervention group and a talk about diabetes education for the control group. The intention to treat (ITT) method will be used for statistical analysis. Statistical analysis will be done using IBM SPSS Statistics version 27. Descriptive analysis will be conducted to check the error and distribution of data. A histogram with a standard curve will be used for normality checking. Independent-t test will be used to investigate

the difference of diabetes knowledge test between groups. Paired t test will be used pretest and post-test after six months. Generalized Estimating Equation (GEE) will be used to investigate the effectiveness of diabetes knowledge and clinical outcome, adjusted with covariates. Time, group, interaction of time and group variables and covariates will be included in the GEE to understand the changes of knowledge and clinical outcome over the time points between control and intervention group. $P < 0.05$ will be considered statistically significant.

## Ethical considerations and declaration

This study adheres to ethical standards outlined in the Declaration of Helsinki and the Malaysian Good Clinical Practice Guideline. Before initiating any study-related activities, ethical approval will be obtained from the Medical Research & Ethics Committee (MREC) and National Medical Research Registry (NMRR). Our research study was registered under Thai Clinical Trials Registry (identification number: TCTR20230321005). The clinical trial registration ensure transparency, accountability and integrity by documenting key trail details publicly before the study begin. It prevents selective reporting and avoid unnecessary duplication of research. All investigators and the study site team must comply with data protection legislation, ensuring the confidentiality of personal information. Participants will be assigned study identification numbers to maintain anonymity.

Access to collated participant data will be restricted to individuals from the research team treating the participants, sponsor representatives, and representatives of regulatory authorities. Paper copies of collected information will be securely stored at Universiti Putra Malaysia. At the same time, electronic data, including audio recordings and transcripts, will be anonymised and accessible only with a secure password. Data will be retained for 5 years and then destroyed.

During recruitment, participants will receive patient information sheets in English or Malay based on preference, with a one-week ample period for informed decision-making. The principal investigator can be contacted if any clarifications are needed. Published results will not include identifiable personal data, and participant identities will be concealed during the presentation of study results. Participants who wish to withdraw from the study can withdraw anytime by informing the principal investigator.

## Status and timeline of study

The study's timeline is shown in (Table 1). At this point, no data collection has taken place.

## Discussion

Malaysia holds the highest diabetes rate in the Western Pacific region and ranks among the highest globally, contributing to an annual economic cost of approximately 600 million United States dollars. Diabetes is expected to affect 7 million Malaysian adults aged 18 and older by 2025, posing a major public health risk with a diabetes prevalence of 31.3%. [25] The effect of improving self-care in diabetic patients on disease mortality and morbidity is significant. [26] Therefore, our study is to investigate the effectiveness of web-based education videos that include tailored information on knowledge to educate type 2 diabetes patients in primary care. The findings from our study will enrich the existing literature on diabetes interventions by showing the effectiveness of web-based educational videos on knowledge of diabetes mellitus and diabetic self-care. This Randomized Controlled Trial (RCT) is an approach to improve recruitment strategies, training materials, and the implementation protocol for a larger cluster RCT in the future. The findings from this study will be shared with academics and the community of our sample population, fostering sustained improvements in diabetes outcomes and promoting health equity.

This study intends to use a new approach towards helping healthcare providers explain to patients regarding type 2 diabetes mellitus. In this modern era, people are more invested towards social media and the internet platform has become the main source of information. Therefore with the development of web-based educational videos, people can enhance their knowledge of type 2 diabetes mellitus effectively. As a result, they will better understand their treatment

**Table 1. Timeline of the study.**

| Project activities | 2022 | | | | | | | | | | | | 2023 | | | | | | | | | | | | 2024 | | 2025 | |
|---|---|---|---|---|---|---|---|---|---|---|---|---|---|---|---|---|---|---|---|---|---|---|---|---|---|---|---|---|
| | J | F | M | A | M | J | J | A | S | O | N | D | J | F | M | A | M | J | J | A | S | O | N | D | Jan-Jun | July-Dec | Jan-Jun (S1) | July-Dec (S2) |
| 1. Project implementation plan write up | | | | | | � | | | | | | | | | | | | | | | | | | | | | | |
| 2. Development of web-based educational videos | | | | | | | | ▪ | ▪ | ▪ | ▪ | ▪ | ▪ | | | | | | | | | | | | | | | |
| 3. Ethic approval and permission from authority | | | | | | | | | | | ▪ | ▪ | ▪ | | | | | | | | | | | | | | | |
| 4. Data collection | | | | | | | | | | | | | | | | | | | | | | | ▪ | ▪ | ▪ | | | |
| 5. Data cleaning, entry and analysis | | | | | | | | | | | | | | | | | | | | | | | | | | ▪ | ▪ | |
| 6. Write up and finalisation | | | | | | | | | | | | | | | | | | | | | | | | | | | | ▪ |

| Project (Milestone) | 2022 | | | | | | | | | | | | 2023 | | | | | | | | | | | | 2024 | | 2025 | |
|---|---|---|---|---|---|---|---|---|---|---|---|---|---|---|---|---|---|---|---|---|---|---|---|---|---|---|---|---|
| | J | F | M | A | M | J | J | A | S | O | N | D | J | F | M | A | M | J | J | A | S | O | N | D | Jan-Jun | July-Dec | Jan-Jun (S1) | July-Dec (S2) |
| 1. Complete Proposal | | | | | | | ✦ | | | | | | | | | | | | | | | | | | | | | |
| 2. Complete Pre-Test | | | | | | | | | | | | ✦ | | | | | | | | | | | | | | | | |
| 3. Complete Data Collection & Analysis | | | | | | | | | | | | | | | | | | | | | | | | | ✦ | | ✦ | |
| 5. Complete dissertation writing | | | | | | | | | | | | | | | | | | | | | | | | | ✦ | | ✦ | |
| 6. Complete Seminar | | | | | | | | | | | | | | | | | | | | | | | | | | | ✦ | |
| 7. Project completion | | | | | | | | | | | | | | | | | | | | | | | | | | | | ✦ |

options and participate in patient-centred care. Understanding the disease will allow them to better focus on their health and ease the burden of healthcare providers on managing their disease. The web-based approach will be our initial step towards making our country a well-managed diabetes centre.

The study also intends to use a multi-language approach to address the challenge of the multicultural and multilingual population in Malaysia, especially in terms of delivery of healthcare education. Deliver educational video in their mother tongue is hopefully able to reduce the misunderstanding of communication between healthcare providers and the patients and make patient more easy to engage with the educational content.

Our study has certain limitations. Firstly, the sample is drawn from patients attending Klinik Kesihatan Cheras Baru, which may not represent the broader population of type 2 diabetes patients. Different socioeconomic backgrounds may reflect on our web-based approach as some may not have access to proper internet connection or smartphones. Second, the participants are possibly exposed to the Hawthorne Effect or observer bias. When patients are constantly reminded to watch a video before being tested on their knowledge, they tend to perform better than they would otherwise. In real-life clinical situations, patients aren't usually reminded as often to watch such videos. Previous study demonstrated that forgetfulness is one of the main reasons (88%) for non-compliance with treatment plans among Diabetes Mellitus patients. [27] Eliminating the forgetfulness factor by frequently calling the patient might not reflect the real effect on daily clinical practice. Next, participants may not consistently complete the videos or follow-up appointments as required, affecting the validity.

Other digital health interventions for diabetes self-management include telemedicine consultations. Telemedicine serves as an effective platform for exchanging health information between patients and healthcare providers. However, it has limitations in settings with underdeveloped infrastructure and a lack of technical expertise. Bureaucratic challenges may arise due to a lack of personal contact, potentially leading to a breakdown in the relationship between healthcare providers and patients. Furthermore, a previous study conducted in the Netherlands [28] utilized mobile health applications and demonstrated good clinical outcomes in terms of diabetes control. In this study, for patients with minimal digital skills, providing instruction and coaching on app usage is crucial as participants were required to familiarize themselves with the app and subsequently monitor and record data, including food intake and physical activity.

In our study, there will be well-trained research assistants to assist and teach the participants how to assess and use the web-based online education videos with a QR code. These research assistants will follow up their progress every 4 weeks as well.

Governments and healthcare organizations could consider integrating educational videos into diabetes management programs, recognizing their cost-effectiveness and scalability. Additionally, our findings could serve as a foundation for developing standardized, evidence-based video content tailored to the cultural and linguistic needs of Malaysian populations. These videos could act as a low-cost solution to address the education gap in areas with limited healthcare worker availability. Furthermore, healthcare workers in resource-limited settings could use these videos to enhance their efforts, ensuring consistent messaging and patient education.

## Supporting information

**S1. Model consent form.**
(ZIP)

## Acknowledgments

The authors acknowledge the Department of Family Medicine team for their contributions to drafting this study protocol.

## Author contributions

**Conceptualization:** Hui Zhu Thew, Poh Ying Lim.

**Data curation:** Hui Zhu Thew.

**Formal analysis:** Hui Zhu Thew.

**Funding acquisition:** Hui Zhu Thew.

**Investigation:** Hui Zhu Thew.

**Methodology:** Hui Zhu Thew, Poh Ying Lim.

**Project administration:** Hui Zhu Thew, Ai Theng Cheong, Sazlina Shariff Ghazali, Aneesa Abdul Rashid, Izwan Zuhrin bin Abdul Malek, Yoke Mun Chan.

**Resources:** Hui Zhu Thew, Yan Bin Fong, Chee Han Ng, Ja Shen Choy, Shen Horng Chong.

**Software:** Hui Zhu Thew, Ai Theng Cheong, Sazlina Shariff Ghazali, Aneesa Abdul Rashid, Izwan Zuhrin bin Abdul Malek, Yoke Mun Chan, Yan Bin Fong, Chee Han Ng, Ja Shen Choy, Shen Horng Chong.

**Supervision:** Hui Zhu Thew, Yan Bin Fong, Chee Han Ng, Ja Shen Choy, Shen Horng Chong.

**Validation:** Hui Zhu Thew, Ping Foo Wong, Yong Kui Choon, Adam Firdaus bin Dahlan, Chitra Suluraju, Nadiah Othman, Yong Wai Leng, Yan Bin Fong, Chee Han Ng, Ja Shen Choy, Shen Horng Chong.

**Visualization:** Hui Zhu Thew, Ping Foo Wong, Yong Kui Choon, Adam Firdaus bin Dahlan, Chitra Suluraju, Nadiah Othman, Yong Wai Leng.

**Writing – original draft:** Hui Zhu Thew.

**Writing – review & editing:** Hui Zhu Thew, Ping Foo Wong, Yong Kui Choon, Adam Firdaus bin Dahlan, Chitra Suluraju, Nadiah Othman, Yong Wai Leng, Yan Bin Fong, Chee Han Ng, Ja Shen Choy, Shen Horng Chong.

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
