## [Decision Letter · Decision Letter 0]

30 Jul 2024

PONE-D-24-14861To develop online platform and determine its effectiveness in ENHANCING DIABetes knowledge among diabetes patients in primary CARE clinic (Enhancing-Diab-Care Study): Study protocolPLOS ONE

Dear Dr. Thew,

Thank you for submitting your manuscript to PLOS ONE. After careful consideration, we feel that it has merit but does not fully meet PLOS ONE’s publication criteria as it currently stands. Therefore, we invite you to submit a revised version of the manuscript that addresses the points raised during the review process.

We look forward to receiving your revised manuscript.

Kind regards,

Muhammad Junaid Farrukh

Academic Editor

PLOS ONE

Journal Requirements:

3. Please include your tables as part of your main manuscript and remove the individual files. Please note that supplementary tables (should remain/ be uploaded) as separate "supporting information" files

4. Please upload a copy of Supporting Information Figure/Table/etc. S1 Fig 1-2, S3 Table 1 and S4. Model consent form  which you refer to in your text on page 15.

Reviewers' comments:

Reviewer's Responses to Questions

**Comments to the Author**

1. Does the manuscript provide a valid rationale for the proposed study, with clearly identified and justified research questions?

Reviewer #1: Yes

Reviewer #2: Yes

2. Is the protocol technically sound and planned in a manner that will lead to a meaningful outcome and allow testing the stated hypotheses?

Reviewer #1: Yes

Reviewer #2: Yes

3. Is the methodology feasible and described in sufficient detail to allow the work to be replicable?

Reviewer #1: Yes

Reviewer #2: Yes

4. Have the authors described where all data underlying the findings will be made available when the study is complete?

Reviewer #1: Yes

Reviewer #2: Yes

5. Is the manuscript presented in an intelligible fashion and written in standard English?

Reviewer #1: Yes

Reviewer #2: Yes

6. Review Comments to the Author

You may also provide optional suggestions and comments to authors that they might find helpful in planning their study.

Reviewer #1: The authors aim to conduct a blocked randomised control trial to evaluate whether an online education video platform can enhance T2D patients’ knowledge about T2D. The intervention group will have additional routine diabetic follow-ups at 3 months and 6 months compared to the control group. The difference between the two arms and the effectiveness of diabetes knowledge will be assessed.

1. Some editing is needed. For example, “creating and creating educational materials…” in the introduction.

2. During the review and finalization phase, all patients will be invited to participate in the review and finalization phase. Will these patients be the same or overlap with those for the later trials? If so, will this create a biased result?

3. Study design section claims data collector will remain blinded for group assignment. However, data collection section mentioned that the research assistant will approach patients on their follow-up visit. Please clarify these two sentences which somewhat conflicts with each other.

4. Will the education level or duration of T2D be considered in study design or analysis?

5. The inclusion/exclusion criteria contains “able to understand Malay”. Considering several languages will be designed, why to have this criterion?

6. The self-administered questionnaires will be in the Malay language. But the web-based education videos will be designed in English first. Why use a different language as default?

Reviewer #2: Authors to clarify study site. Kindly refer to edited attachment for language and grammar recommendations

7. PLOS authors have the option to publish the peer review history of their article (what does this mean? ). If published, this will include your full peer review and any attached files.

**Do you want your identity to be public for this peer review?** For information about this choice, including consent withdrawal, please see our Privacy Policy .

Reviewer #1: No

Reviewer #2: No

---

## [Author Response · Author response to Decision Letter 1]

2 Sep 2024

Thank you for the review, we had made corresponding changes according to the previous email. We have hereby attached a file named "response to reviewer" to address all the comments from the reviewer.

For reviewer 1:

1) The authors aim to conduct a blocked randomised control trial to evaluate whether an online education video platform can enhance T2D patients’ knowledge about T2D. The intervention group will have additional routine diabetic follow-ups at 3 months and 6 months compared to the control group. The difference between the two arms and the effectiveness of diabetes knowledge will be assessed.

Thank you for your feedback. We have made amendments as per mentioned in the section: Data collection

“After three months of the intervention, both the intervention and control group participants will return to the clinics for follow-up.”

And

“Six months from the baseline, the intervention and control group participants will return to the clinics for follow-up.”

Both the control and intervention groups will not receive any additional diabetic follow-up. Patients in both the intervention and control groups will have follow-up appointments at 3 months and 6 months. The only difference is that patients in the intervention group will be provided with a diabetes care educational video and will be reminded to watch it. Routine follow-up and diabetes care provided in the clinic will be similar for both groups to avoid bias in the results.

2) Some editing is needed. For example, “creating and creating educational materials…” in the introduction.

Thank you for the comment. The corresponding error has been rectified and corrected in which part, copy paste here to show them and highlight in the manuscript

“Utilising the media in publicity campaigns, creating educational materials, and educating people with diabetes and their families should be part of this effort.”

3) During the review and finalization phase, all patients will be invited to participate in the review and finalization phase. Will these patients be the same or overlap with those for the later trials? If so, will this create a biased result?

Thank you for sharing your comment. These were two different groups of patients, and the patients were different and did not overlap with the trial patients. We also added in the manuscript for clarity in which part and highlight in the manuscripts under the section “review and finalization” as follow: “ These were two different groups of patients, and the patients were different and did not overlap with the trial patients. They will review the scripts and complete a questionnaire based on the 12 video topics. ”

4) Study design section claims data collector will remain blinded for group assignment. However, data collection section mentioned that the research assistant will approach patients on their follow-up visit. Please clarify these two sentences which somewhat conflicts with each other.

Thank you for pointing out the issue. Under the “study design” section, we have made the corresponding corrections under section “study design” to address this conflict “The research assistant will undergo training to adhere to standardised protocols during data collection.”

5) Will the education level or duration of T2D be considered in study design or analysis?

Thank you for your comment. Our questionnaire includes questions about both the educational level and the duration of T2DM diagnosis. The educational level is addressed in the "Latar Belakang Sosio Demografi (demography and social background)" section, question number 4, and the duration of T2DM diagnosis is covered in the "Maklumat perubatan (medical history)" section, question number 5.

6) The inclusion/exclusion criteria contains “able to understand Malay”. Considering several languages will be designed, why to have this criterion?

Thank you for your comment. We include participants who are able to understand Malay because our questionnaire is administered in Malay because Malay is the official language in Malaysia. The main reason for creating the video in four languages (Malay, Chinese, Tamil, and English) is due to the multiracial background of the patients. Learning in one's own mother tongue has been proven to be more effective, as demonstrated in many studies.

7) The self-administered questionnaires will be in the Malay language. But the web-based education videos will be designed in English first. Why use a different language as default?

The main video is in English, and the script is based on the most recent Clinical Practice Guidelines from Malaysia, alongside references from the American Association of Clinical Endocrinology (AACE) and the European Association for the Study of Diabetes (EASD). The information has been summarized and written in simpler terms for easier understanding. The expert panel will review the scripts and complete a questionnaire based on the 12 video topics. Each topic will be evaluated on two items: relevance and clarity, using a 3-point scale where 1 indicates 'not relevant' or 'not clear,' 2 indicates 'somewhat relevant' or 'somewhat clear,' and 3 indicates 'highly relevant' or 'highly clear.' The content and validity of each video were acceptable, with a score of 0.8. The English version was translated into Malay, Chinese, and Tamil languages, and a pilot study was conducted. A sample of 5 respondents for each language was selected. Some minor grammatical errors and difficult terminology were identified and addressed accordingly. A reliability test was conducted again for the Malay, Chinese, and Tamil languages, and the results were acceptable.

The questionnaire was administered in Malay because Malay is the official language in Malaysia. The main reason for creating the video in four languages (Malay, Chinese, Tamil, and English) is due to the multiracial background of the patients. Learning in one's own mother tongue has been proven to be more effective. Coughlin, et al (2020).

For Reviewer 2:

Authors to clarify study site. Kindly refer to the edited attachment for language and grammar recommendations

Thank you for the comment. The corresponding grammar and language error have been rectified and corrected.

---

## [Editor Report · Decision Letter 1]

16 Sep 2024

PONE-D-24-14861R1To develop online platform and determine its effectiveness in ENHANCING DIABetes knowledge among diabetes patients in primary CARE clinic (Enhancing-Diab-Care Study): Study protocolPLOS ONE

Dear Dr. Thew, 

Thank you for submitting your manuscript to PLOS ONE. After careful consideration, we feel that it has merit but does not fully meet PLOS ONE’s publication criteria as it currently stands. Therefore, we invite you to submit a revised version of the manuscript that addresses the points raised during the review process. Please submit your revised manuscript by Oct 31 2024 11:59PM. If you will need more time than this to complete your revisions, please reply to this message or contact the journal office at plosone@plos.org . Please include the following items when submitting your revised manuscript:

We look forward to receiving your revised manuscript.

Kind regards,

Muhammad Junaid Farrukh

Academic Editor

PLOS ONE

Journal Requirements:

**Additional Editor Comments:**

Dear authors, thank you for doing extensive revisions. now the manuscript looks more comprehensive and detailed.

however, i would like to request you to please include the following 2 items in your manuscript.

1) sample size calculation:

The sample size was calculated based on two mean groups formula. kindly include/insert the formula and show the breakdown of the calculation for easy understand of the readers.

2) web-based education video

please elaborate how the videos were developed? what information sources were used? was it recorded from scratch or some online videos were used? any copyright issue? 

Kindly upload the videos in google drive or share the link so we can verify them.

thank you

---

## [Author Response · Author response to Decision Letter 2]

16 Oct 2024

1) sample size calculation:

The sample size was calculated based on two mean groups formula. kindly include/Insert the formula and show the breakdown of the calculation for easy understand of the readers.

Thank you for your feedback. We have made amendments as per mentioned in the section: Sample size

Two mean formula is stated as below:

n = (〖2σ〗^2 [z_(1-α/2)+z_(1-β) ]^2)/(μ_1-μ_2 )^2

n sample size estimate

Z1-∝/2 standard error with 95% confidence interval= 1.96

z1−β Standard error associated with 80% power= 0.842

μ_1 mean of knowledge score in intervention group

μ_2 mean of knowledge score in control group

Estimated standard deviation

n = (〖2(2.27)〗^2 [1.96+0.842]^2)/(1)^2

=81 per group

Adjusted with 30% attrition rate = 81/0.7

=116 per group

Total sample size is 116 x 2 =232

2) web-based education video

Please elaborate how the videos were developed?

what information sources were used?

was it recorded from scratch or some online videos were used?

Any copyright issue?

Kindly upload the videos in google drive or share the link so we can verify them.

Thank you for your feedback. We have made amendments as per mentioned in the section: Developmental phase.

“The videos were created from the ground up. All branding logos were carefully designed to respect intellectual property rights, and the graphics and background music were sourced from free, authorized channels to avoid any copyright concerns. The logos were also custom-designed to ensure originality.

Upon completion of recording and editing, each version of the videos in the four languages (English, Malay, Chinese, and Tamil) underwent the copyright process to protect their intellectual property.”

The video were attached in the google drive link provided:

Chinese and English version: https://drive.google.com/drive/folders/1jhgg9pfLSdvEqy3hu2_9jVt-l-Yup4WZ?usp=sharing

Malay and Tamil version: https://drive.google.com/drive/folders/1krcnEwcq_A0oChv6fZ3o5Xss6cdiY3da?usp=sharing

---

## [Decision Letter · Decision Letter 2]

20 Dec 2024

PONE-D-24-14861R2To develop online platform and determine its effectiveness in ENHANCING DIABetes knowledge among diabetes patients in primary CARE clinic (Enhancing-Diab-Care Study): Study protocolPLOS ONE

Dear Dr. Thew,

Thank you for submitting your manuscript to PLOS ONE. After careful consideration, we feel that it has merit but does not fully meet PLOS ONE’s publication criteria as it currently stands. Therefore, we invite you to submit a revised version of the manuscript that addresses the points raised during the review process.

 Please submit your revised manuscript by Feb 03 2025 11:59PM. If you will need more time than this to complete your revisions, please reply to this message or contact the journal office at plosone@plos.org . Please include the following items when submitting your revised manuscript:

We look forward to receiving your revised manuscript.

Kind regards,

Muhammad Junaid Farrukh

Academic Editor

PLOS ONE

Reviewers' comments:

Reviewer's Responses to Questions

**Comments to the Author**

1. Does the manuscript provide a valid rationale for the proposed study, with clearly identified and justified research questions?

Reviewer #3: Partly

Reviewer #4: Yes

Reviewer #5: Yes

Reviewer #6: Yes

2. Is the protocol technically sound and planned in a manner that will lead to a meaningful outcome and allow testing the stated hypotheses?

Reviewer #3: Partly

Reviewer #4: Yes

Reviewer #5: Partly

Reviewer #6: Yes

3. Is the methodology feasible and described in sufficient detail to allow the work to be replicable?

Reviewer #3: Yes

Reviewer #4: Yes

Reviewer #5: No

Reviewer #6: Yes

4. Have the authors described where all data underlying the findings will be made available when the study is complete?

Reviewer #3: Yes

Reviewer #4: Yes

Reviewer #5: Yes

Reviewer #6: No

5. Is the manuscript presented in an intelligible fashion and written in standard English?

Reviewer #3: Yes

Reviewer #4: Yes

Reviewer #5: No

Reviewer #6: Yes

6. Review Comments to the Author

You may also provide optional suggestions and comments to authors that they might find helpful in planning their study.

Reviewer #3: Dear Authors,

Overall the study may offer valuable inputs for the improvement of T2DM healthcare outcomes. The introduction part was clear and concise however some minor amendments may be required for the abstract and methods part.

Abstract:

Line 3 of abstract - 'disease.. Maintaining' - double full-stop

Line 4-'The COVID-19 pandemic showed a shift of attention from non-communicable to communicable diseases' - this statement is controversial as this statement seemed to be generalised. Is this accurate for all healthcare institutions, healthcare providers and patients?

2nd paragraph- Rephrase this '232 type 2 diabetes mellitus patients will consent and voluntarily participate'. Since data has not been collected, may write as the minimum sample size calculated is 232 T2DM patients. Patients will participate on a voluntary basis where informed consent will be requested prior to data collection (rephrase necessarily).

2nd paragraph - Suggest to rephrase 'A routine diabetic follow-up......' to - Patients undergoing diabetic follow-up will be divided into two groups where only patients within the intervention group will receive diabetic care video.

3rd paragraph - 'Baseline questionnaires will be self-administered with their clinical profile to assess

diabetes self-care knowledge and empowerment.' Suggest to change to 'Patients clinical profile such as (i.e give examples) were collected to assess diabetes self-care knowledge and empowerment using self-administered questionnaire.'

3rd paragraph - ' reminded to complete the educational video at the 3-month and 6-month follow-ups, and the aforementioned parameters will be reassessed'- Are the patients required to revisit the videos at 3 months and 6 months? For those who have completed viewing the videos earlier, researchers will have to collect their clinical parameters only?

Study Population - Will there be any difference in terms of knowledge between walk in patients with patients followed-up (considering if they are being followed-up in another institution previously which offered them very good patient education)? relevance of duration for data collection?

Development phase - State CPG which will be used? Standardise the tenses used 'The information has been summarized.In this phase, a set of English-version online education videos will be developed. They are presented in a simple, short,'. i.e will, has been, are presented. Similarly 'The English version was translated into Malay, Chinese, and Tamil languages, and a pilot study was conducted. A sample of 5 respondents for each language was selected. Some minor grammatical errors and difficult terminology were identified and addressed accordingly. A reliability test was conducted again for the Malay, Chinese, and Tamil languages, and the results were acceptable'. I think these sentences should be written as future tense.

Review and finalisation phase- Do you think would it be better to include certified translators as expert panels?

Data collection -'no further treatment will be administered after the study is completed'. Can you explain what do you mean by no further treatment administered?

Study instrument- Why dont you consider translating the questionnaire into multiple languages? Are all different stages of patients going to view the same set of videos? Do you think this is suitable?

Overall, this study may offer some benefits in improving T2DM patients' healthcare outcomes. However, the extent of information exposed to patients must be deliberately scrutinized to ensure the correct input and correct amount of information were given at the appropriate time in ensuring effectiveness of patient education method.

Reviewer #4: The authors have addressed all the comments and concerns raised during the peer review process. In my opinion the manuscript is acceptable for publication.

Reviewer #5: Overall Comments

• The study is interesting and has potential; however, the protocol requires significant improvement in its writing.

• Please standardize the citation style. Some citations are placed before the period, while others are after.

• Ensure consistent spacing after periods throughout the document.

Methods

• The protocol outlines several phases. Please provide detailed methods for each phase.

Video

• Specify who verified or validated the video and indicate how many individuals were involved.

• Clarify who was involved in the translation and validation of the video into Malay, Chinese, and Indian languages.

• Include a reference for the scale used to assess content validity.

Diabetes Education

• Describe the content of the diabetes education in detail.

• Explain how the education will be conducted and identify the individuals involved in delivering it.

Knowledge on Diabetes

• Provide details about the questionnaires used, including their validity, language, scale, scoring, and domains.

Randomization and Bias Minimization

• Elaborate on the block randomization process. Specify the number of blocks and who will manage the randomization.

• Address how bias will be minimized, given that participants are from the same clinic.

Statistical Analyses

• Elaborate on the use of the “Generalised Estimating Equation.”

• Are you applying generalized linear regression to identify the factors?

Discussion

As this is a study protocol, a discussion section is not required. However, it would be helpful to include a study flowchart and the expected results.

Reviewer #6: Elaborate on the novelty of the approach, especially how the multi-language videos address challenges in diverse populations.

Provide more details on how the Generalized Estimating Equations (GEE) will account for covariates.

Clarify the criteria for significant improvement in diabetes knowledge and clinical outcomes.

Compare the proposed intervention with other digital health solutions for diabetes self-management in recent studies.

Discuss how findings could influence policies or practices in resource-limited settings.

Expand on the challenges of implementing such interventions in regions with limited internet access or digital literacy.

Consider including the potential for bias due to the Hawthorne effect or participant attrition.

Add a schematic overview of the intervention and assessment process for clarity.

7. PLOS authors have the option to publish the peer review history of their article (what does this mean? ). If published, this will include your full peer review and any attached files.

**Do you want your identity to be public for this peer review?** For information about this choice, including consent withdrawal, please see our Privacy Policy .

Reviewer #3: **Yes: ** Aina Yazrin Ali Nasiruddin

Reviewer #4: **Yes: ** Dr A Vijaya Bhaskar Reddy

Reviewer #5: No

Reviewer #6: No

---

## [Author Response · Author response to Decision Letter 3]

8 Feb 2025

Thank you for the review, we had made corresponding changes according to the previous email. We have hereby attached a file named "response to reviewer" to address all the comments from the reviewer.

---

## [Decision Letter · Decision Letter 3]

3 Apr 2025

To develop online platform and determine its effectiveness in ENHANCING DIABetes knowledge among diabetes patients in primary CARE clinic (Enhancing-Diab-Care Study): Study protocol

PONE-D-24-14861R3

Dear Hui Zhu Thew

We’re pleased to inform you that your manuscript has been judged scientifically suitable for publication and will be formally accepted for publication once it meets all outstanding technical requirements.

Kind regards,

Muhammad Junaid Farrukh

Academic Editor

PLOS ONE

Additional Editor Comments (optional):

Reviewers' comments:

Reviewer's Responses to Questions

**Comments to the Author**

1. Does the manuscript provide a valid rationale for the proposed study, with clearly identified and justified research questions?

Reviewer #1: Yes

Reviewer #7: Partly

2. Is the protocol technically sound and planned in a manner that will lead to a meaningful outcome and allow testing the stated hypotheses?

Reviewer #1: Yes

Reviewer #7: Yes

3. Is the methodology feasible and described in sufficient detail to allow the work to be replicable?

Reviewer #1: Yes

Reviewer #7: Yes

4. Have the authors described where all data underlying the findings will be made available when the study is complete?

Reviewer #1: Yes

Reviewer #7: Yes

5. Is the manuscript presented in an intelligible fashion and written in standard English?

Reviewer #1: Yes

Reviewer #7: Yes

6. Review Comments to the Author

You may also provide optional suggestions and comments to authors that they might find helpful in planning their study.

Reviewer #1: Thanks for addressing all the raised concerns in previous submission. I have no further comments on this revised version.

Reviewer #7: KAP terminology is used for knowledge attitude and practice. ?? Awareness

Introduction 3rd para- mention NCD is bracket at first instance of mention non communicable disease

Introduction last para- tailored information on knowledge to educate - the line implicates the information is on how to educate rather than educating.

7. PLOS authors have the option to publish the peer review history of their article (what does this mean? ). If published, this will include your full peer review and any attached files.

**Do you want your identity to be public for this peer review?** For information about this choice, including consent withdrawal, please see our Privacy Policy .

Reviewer #1: No

Reviewer #7: No

---

## [Editor Report · Acceptance letter]

PONE-D-24-14861R3

PLOS ONE

Dear Dr. Thew,

I'm pleased to inform you that your manuscript has been deemed suitable for publication in PLOS ONE. Congratulations! Your manuscript is now being handed over to our production team.

Kind regards,

on behalf of

Dr. Muhammad Junaid Farrukh

Academic Editor

PLOS ONE